# Brain health index as a predictor of possible vascular dementia in the Mexican health and aging study 2012–2015

Sara G. Aguilar-Navarro[1,2]*, Sara G. Yeverino-Castro[2,3], Silvia Mejía-Arango[4],
Rogelio Moctezuma[1], Teresa Juárez-Cedillo[5], Alberto José Mimenza-Alvarado[1,2]

1 Deparment of Geriatric Medicine & Neurology Fellowship, Instituto Nacional de Ciencias Médicas y
Nutrición Salvador Zubirán, Tlalpan, Mexico City, Mexico, 2 Department of Geriatric Medicine, Instituto
Nacional de Ciencias Médicas y Nutrición Salvador Zubirán, Tlalpan, Mexico City, Mexico, 3 CHRISTUS
Center of Excellence and Innovation, San Pedro Garza García, Nuevo León, México, 4 Department of
Population Studies, El Colegio de la Frontera Norte, Tijuana, Baja California, México, 5 Epidemiologic and
Health Service Research Unit, Aging Area, Mexican Institute of Social Security, National Medical Center
Century XXI, Mexico City, Mexico

☯ These authors contributed equally to this work.
* sara.aguilarn@incmnsz.mx

journal.pone.0304234

Facultad de Estudios Superiores Zaragoza:
Universidad Nacional Autonoma de Mexico
Facultad de Estudios Superiores Zaragoza, MEXICO

## Abstract

To determine the burden of disease among subjects at risk of developing stroke or dementia, brain health indexes (BHI) tend to rely on anatomical features. Recent definitions emphasize the need of a broader perspective that encompasses cardiovascular risk factors (CVRFS) and lifestyle components which can be considered partial contributors to optimal brain health. In this study, we aimed to establish the association and risk detected by a Brain Health Index and the risk of possible vascular dementia (PVD) using data from the Mexican Health and Aging Study (MHAS) 2012–2015. The MHAS is a longitudinal study of adults aged ≥ 50 years. We analyzed the data obtained between 2012 and 2015. CVRFS included in the index were diabetes mellitus, hypertension, myocardial infarction, depression, obesity, physical inactivity, and smoking history. A PVD diagnosis was established when scores in the Cross-Cultural Cognitive Examination were below reference norms and limitations in ≥1 instrumental activities of daily living and a history of stroke were present. A multinomial regression model was developed to determine the association between BHI scores and PVD. In 2015, 75 PVD cases were identified. Mean age was 67.1 ±13.2 years, 35.8% were female, and the mean educational level was 5.8 ±5.5 years. In cases with a higher score in the BHI, the model revealed a hazards ratio of 1.63 (95% CI: 1.63–1.64, p< 0.001) for PVD. In this longitudinal study, with the use of a feasible multifactorial BHI in the Mexican population, a greater score was associated with a 1.63-fold risk of developing PVD during the 3-year follow-up, while the risk for stroke was 1.75. This index could potentially be used to predict the risk of PVD in adults with modifiable CVRFS.

**Data Availability Statement:** All relevant data are within the manuscript and its Supporting Information files. Data are available from The Mexican Health and Aging Study (MHAS) from www.mhasweb.org, which are of public use".

**Funding:** NO: The funders had no role in study design, data collection and analysis, decision to publish, or preparation of the manuscript.

**Competing interests:** The authors have declared that no competing interests exist.

## Introduction

Vascular cognitive impairment (VCI) refers to various cerebrovascular pathologies that contribute to cognitive impairment [1]. Vascular dementia (VaD) is the second leading cause of dementia accounting for up to 15–20% of dementia cases [2]. This condition develops in up to 15–30% of patients within 3 months of a stroke event [2]. When compared to Alzheimer's disease (AD), VaD patients have both a higher cardiovascular burden and a greater level of disability, which in turn leads to increased health-related costs [3]. Annually, the global cost of dementia alone has been estimated to be around a trillion US dollars [4].

There is limited information concerning VCI in countries such as Mexico. The Fogarty study analyzed 110 Mexican subjects with a first-ever stroke and found a 3-month VaD prevalence of 12% [5]. Even when the gold standard diagnostic procedure was not available, authors estimated a prevalence of 0.6% (95% CI: 0.5–0.8) of possible vascular dementia (PVD) with an incidence of 2.2 cases (95% CI: 1.7–2.6) per 1000 person-years, based on data from the Mexican Health and Aging Study (MHAS) [6]. Possible mild or major VCI (VaD) is a feasible diagnosis when neuroimaging is not available and clinically significant cognitive deficits in at least one cognitive domain with or without functional dependence are present [7].

Brain health is defined as the capacity to function adaptively in the environment, with recent definitions emphasizing the need for a broader perspective and not only referring to the mere absence of disease such as stroke or AD [8]. Various efforts have been made to quantify brain health and dementia risk and establish prevention goals [9], given that in the absence of effective treatment of the disease spectrum, cardiovascular risk factors (CVRFSs) are a potential target for the prevention of all-cause dementia [10]. In addition, since most CVRFSs are considered chronic conditions, the development of tools to measure the impact of their coexistence rather than their individual contribution, could be helpful to determine relevant outcomes. For instance, the American Heart Association's (AHA) Life's Simple 7 is one of the tools serving this purpose [11]. Among many other contributing factors, cardiovascular disease has been previously associated with different pathological dementia models [12]. Thus, cardiovascular risk factors (CVRFS), especially those included in the AHA Life's Simple 7 Index, can be considered partial contributors to optimal brain health [8]. Furthermore, the Lifestyle for Brain Health (LIBRA) index, that assesses modifiable CVRFS, has been proven useful in identifying a higher risk for cognitive impairment [13, 14].

However, to determine the burden of disease among subjects at risk of developing stroke or dementia, brain health indexes (BHI) tend to rely on anatomical features that quantify and detect, with neuroimaging techniques such as magnetic resonance imaging (MRI), the presence of cerebral small-vessel disease and brain atrophy [15].

In a survey that evaluated availability, accessibility, and affordability of neuroimaging tests among low-middle income countries, authors found that one of the least most available tests was the MRI [16]. Furthermore, only a small proportion of the surveyed countries had public health care systems that could provide this resource in less than 48 hours, and lastly, it was one of the least affordable in low-income populations [16]. Moreover, given that some authors find it essential to include multi-dimensional components that capture brain health capacity in a more holistic approach, other indexes have also considered daily life activities such as sleep patterns and social engagement while incorporating metabolic equivalents of fitness such as quality of life, mood and depression, and well-being [17].

Besides identifying preventable CVRFS as essential components of brain health, cohort studies are needed to demonstrate the use of accessible holistic screening tools that assess the risk of vascular cognitive impairment in the Mexican population, where information on this

topic is limited. This study aimed to establish the association and risk detected by the Brain Health Index as a predictor of PVD in the MHAS 2012–2015.

## Materials and methods

### Population

This observational cohort study included participants from the MHAS, a national representative survey of Mexican subjects aged ≥50 years, with four follow-up waves completed (2003, 2012, 2015, 2018) since its inception in 2001. The MHAS was designed to evaluate the impact of disease on health, function, and mortality in urban and rural zones in Mexico, by focusing on disease burden and disability [18]. Utilizing data obtained between October 2012 and December 2015, a BHI was constructed using several CVRFSs included in the MHAS, to establish its predictor qualities associated to PVD and other conditions.

This manuscript was submitted to the Ethics and Research Committee of the National Institute of Medical Sciences and Nutrition Salvador Zubiran (INCMNSZ for its acronym in Spanish). The MHAS protocol review and ethical approvals were performed by the Institutional Review Board of the University of Texas Medical Branch, the National Institute of Statistics and Geography (INEGI for its acronym in Spanish) and the National Institute of Public Health (INSP for its acronym in Spanish) in Mexico. MHAS data files and documentation are of public use and are available at www.mhasweb.org.

Fig 1 shows the subject selection flowchart in the 2012–2015 waves. Of the initial 14,890 individuals in the sample in 2012, we identified 14,153 cognitively unimpaired (CU) individuals. In the 2012 sample, 12,427 participants had an available cognitive assessment at the time of the 2015 follow-up cutoff. We identified 11,540 CU participants, 513 of whom now fulfilled new-onset dementia without stroke diagnostic criteria, seventy-five had new-onset PVD, and 299 had a history of stroke without dementia.

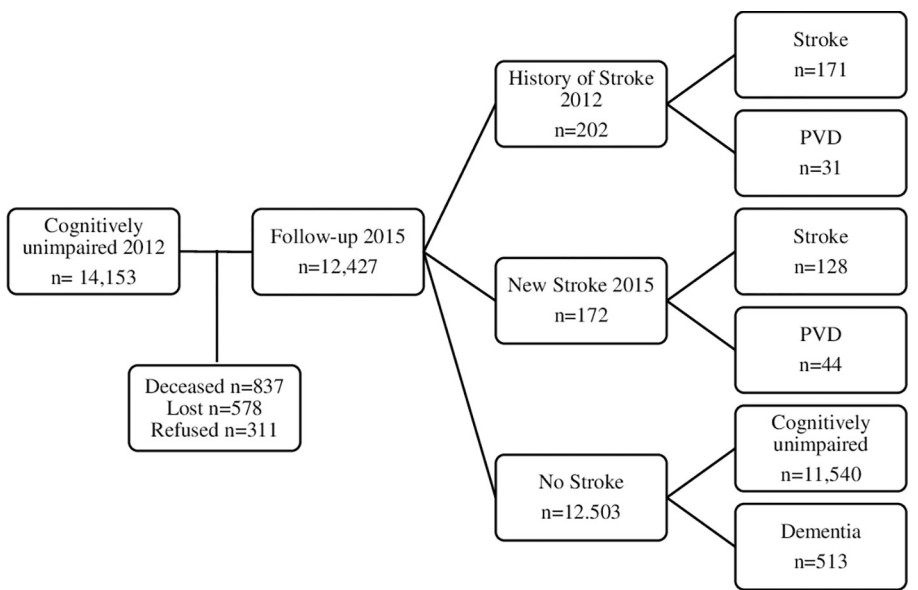

PVD: Possible vascular dementia

**Fig 1. Flowchart of sample selection 2012–2015.** PVD: Possible vascular dementia.

## Cognitive assessment

An adaptation of the Cross-Cultural Cognitive Examination (CCCE) was used for self-respondents, to measure individual performance in eight cognitive domains: verbal learning, delayed memory, attention, constructional praxis, visual memory, verbal fluency, orientation, and processing speed, adjusted for age and educational level [19].

An abbreviated version of the Informant Questionnaire on Cognitive Decline in the Elderly (IQCODE) was used for proxy respondents, which consists of a 16-item questionnaire on cognitive decline in older adults, rated on a 5-point scale from one "much improved" to 5 "much worse." An informant who knew about the participant's daily functioning, usually a spouse, an adult child, or a caregiver, assessed the participant's cognitive status compared to how it was two years earlier [20]. Imputed data were used on cognitive performance measures for individuals with missing values using a multivariate, regression-based procedure previously applied by the MHAS team [21].

**Cognitive categories.** For this study, a participant was defined as an individual with dementia without a stroke, if his/her performance scores in at least two cognitive domains in the CCCE were $\geq$1.5 standard deviations (SD) below the mean, based on norms for age and education, and if they referred difficulties performing at least one instrumental activity of daily living (IADL) [22, 23]. The IADLs assessed in the MHAS included the ability to prepare a meal, go shopping, manage money, or take medications (18). For proxy respondents, an IQCODE score $\geq$ 3.4 was considered for diagnosis [20].

Participants were classified as cognitively unimpaired (CU) if their performance in the CCCE was no more than one standard deviation (SD) below norms in all cognitive domains or $\geq$1.5 SD below in only one domain, and no IADL limitations were present. Proxy respondents with a score <3.4 points in the IQCODE were also included in this category.

Participants were considered to have a "history of stroke" based on a yes/no answer to the MHAS question: "Have you ever been diagnosed with a stroke by a doctor?" at baseline and follow-up. To adjust for a possible response bias, participants were also included if they reported any of the following conditions: focal symptoms of stroke (e.g., inability to move extremities, difficulty speaking/eating, difficulty with sight/vision, difficulty thinking/expressing him/herself), if they received rehabilitation therapy or took medications pertaining to stroke management (e.g., aspirin).

The presence of PVD was defined by the combined presence of a positive history of stroke and dementia. A definition of possible mild or major VCI (VaD) is appropriate when neuroimaging is not available and clinically significant cognitive deficits in at least one cognitive domain with or without functional dependence are present [24].

## Brain health index

The BHI used in this study was derived from previous proposals: the AHA Life's Simple 7, the Lifestyle for Brain Health (LIBRA) Index of the Maastricht Study, and the Cardiovascular Risk Factors, Aging, and Dementia (CAIDE) Finnish Study [11, 13, 25].

To construct the BHI, we considered 7 CVRFSs identified in the MHAS questionnaire and measured them with a score between 0 and 7, whereby a higher score implied a greater risk. The selected factors involving comorbidities were derived from the question "Has a medical doctor diagnosed you with diabetes mellitus, hypertension, or myocardial infarction. To determine the level of physical inactivity, the participant had to answer "no" to the question: "On average, during the last two years, have you exercised or been engaged in hard physical work three or more times a week?".

The presence of depressive symptoms, obesity, and smoking history were also considered in the index. Depressive symptoms were considered significant based on a cut-off score ≥5 in the modified version of the Center for Epidemiological Studies-Depression Scale (CES-D) included in the MHAS [26]. Obesity was defined as a body mass index (BMI) ≥25 kg/m2, calculated from self-reported body weight and height. Lastly, history of smoking consisted of a composite score (0–2) of the following questions: "Have you ever smoked cigarettes?" and "Do you smoke cigarettes now?" coded as 1 or 0.

**Statistical analysis.** Frequencies, means, and standard variations were used to describe covariables at baseline and at follow-up. To account for lost data due to the MHAS follow-up waves, cases were weighted and adjusted for non-responders, as previously detailed by Downer et al [27]. To examine differences in covariates between diagnostic groups, test for continuous variables (one-way analysis of variance, ANOVA, followed by *post hoc* Bonferroni correction) and Chi-square test for categorical variables, were used. A multinomial logistic regression model was developed to establish the risk of developing each dependent variable (dementia, history of stroke, and PVD) associated to BHI scores (independent variable); throughout the 3-year follow-up. The model was adjusted for age, sex, and educational level, which have been previously identified as risk factors for cognitive impairment [10].

## Results

Table 1 shows the participants' demographic characteristics in the 2015 follow-up cutoff. In cases considered to have PVD, the individuals´ mean age was 67.1 ± 13.2 years, 35.8 were female, and the mean educational level was 5.81 (SD ± 5.5) years. The PVD group had a statistically significant higher educational level (5.8 ± 5.5 years) when compared to the dementia (3.5 ± 3.9 years) and the stroke (4.0 ± 3.7 years) groups. In the dementia group, the mean age was 74.1 ± 11.4 years and 63.7% were female. The "history of stroke" group had a mean age of 65.5 ± 9.8 and 48% were female. The CU group had a mean age of 64.4 ± 9.1, 55.4% were female, and the mean educational level was 5.8 ± 4.7 years.

Among the different CVRFS included in the index, both hypertension and diabetes myocardial infarction were the most prevalent among participants in the stroke group (78% and 38.8%, respectively), while the frequency in the PVD group was 51% and 16.5%, respectively. When compared to the CU group, depressive symptoms were more frequent among participants in the dementia group (30.6% *vs.* 55.8%, respectively, p<0.001). Physical inactivity was more frequent among participants in the PVD group (87.3%, p <0.001). Smoking history was also more frequent (59.5%) in the PVD group. Obesity was more common among participants with a stroke history and in CU individuals (28.3% and 25% respectively), while the PVD group had the lowest prevalence (4.3%). Lastly, diabetes mellitus was more frequent among participants in the stroke and dementia groups (38.8% and 31.6%, respectively), and less frequent in the PVD group (16.5%).

When analyzing the BHI as a proxy of multimorbidity, the global score was 2.1 ± 1.2 (Table 1). The CU group had at least 2 positive CVRFSs 2.1 ± 1.2, with an increasing score as the diagnostic category change. Participants in the CU group had a statistically significant lower BHI score of 2.1 ± 1.2, when compared to the dementia group (2.6 ± 1.2). Participants with a stroke history had a score of 3.0 ± 1.2 points, while those in the PVD group had a score of 2.1 ± 1.0.

Characteristics of the MHAS sample in 2012 are shown in the Supplementary material (S1 Table). In the 2012 baseline characteristics, the BHI score in CU individuals was 1.9 ± 1.2 points and in participants with a positive stroke history the score was 3.0 ± 1.2. The BHI scores in the PVD and dementia groups were 2.6 ± 0.9 and 2.9 ± 1.2 points, respectively.

**Table 1. Demographic characteristics of MHAS sample 2015 by diagnostic group.**

| Characteristics | Total<br>n = 12,427 | Cognitively unimpaired<br>n = 11,540 | Dementia<br>n = 513 | Stroke<br>n = 299 | Possible Vascular Dementia<br>n = 75 | p-value |
|---|---|---|---|---|---|---|
| Age*<br>Mean (SD) | 64.8 (9.4) | 64.4 (9.1) | 74.1 (11.4) | 65.5 (9.8) | 67.1 (13.2) | < .001 |
| Female (%) | 55.4 | 55.4 | 63.7 | 48.1 | 35.8 | < .001 |
| Education (y)**<br>Mean (SD) | 5.6 (4.7) | 5.8 (4.7) | 3.5 (3.9) | 4.0 (3.7) | 5.8 (5.5) | < .001 |
| Hypertension (%) | 43.8 | 42.5 | 54.7 | 78.0 | 51.0 | < .001 |
| Diabetes (%) | 22.2 | 21.6 | 31.6 | 38.8 | 16.5 | < .001 |
| Myocardial infarction (%) | 3.2 | 2.8 | 5.6 | 16.5 | 6.9 | < .001 |
| Obesity (%) | 24.8 | 25.0 | 20.6 | 28.3 | 4.3 | < .001 |
| Smoking (%) | 41.5 | 41.8 | 30.8 | 37.6 | 59.5 | < .001 |
| Physical inactivity (%) | 60.4 | 59.6 | 74.5 | 71.2 | 87.3 | < .001 |
| Depressive symptoms (%) | 31.7 | 30.6 | 55.8 | 48.3 | 37.4 | < .001 |
| BHI score*<br>Mean (SD) | 2.1 (1.2) | 2.1 (1.2) | 2.6 (1.2) | 3.0 (1.2) | 2.1 (1.0) | < .001 |

y: years, SD: Standard deviation, BHI: Brain Health Index

P-value from ANOVA for continuous variables and Chi-square for categorical variables. All values were weighted and derived from the MHAS sampling weights.

Bonferroni correction

*All comparisons between groups were significant (p < .001)

**Comparisons were significant (p < .001), except for PVD vs CU.

Table 2 shows the multinomial logistic regression model for the analysis of the association between the BHI and the diagnostic categories; the model was adjusted for age, sex, and educational level. For the development of dementia, in cases with a higher score in the BHI, the model revealed a hazards ratio (HR) of 1.25 (95% CI: 1.24–1.26). In the "history of stroke" group, an even greater HR 1.75 (95% CI: 1.75–1.76, p <0.001) for BHI. In the case of PVD, the

**Table 2. Multinomial regression model for BHI, cognitive categories, and stroke in MHAS 2012–2015.**

| Categories | Variables | HR | CI | p-value |
|---|---|---|---|---|
| Dementia | BHI | 1.25 | 1.24–1.26 | < .001 |
| | Education | 0.96 | 0.96–0.97 | < .001 |
| | Age | 1.08 | 1.08–1.09 | < .001 |
| | Sex* | 0.73 | 0.72–0.73 | < .001 |
| Stroke | BHI | 1.75 | 1.75–1.76 | < .001 |
| | Education | 0.90 | 0.90–0.91 | < .001 |
| | Age | 0.92 | 0.91–0.98 | < .001 |
| | Sex* | 1.77 | 1.76–1.78 | < .001 |
| PVD | BHI | 1.63 | 1.63–1.64 | < .001 |
| | Education | 1.03 | 1.03–1.04 | < .001 |
| | Age | 1.01 | 1.01–1.02 | < .001 |
| | Sex* | 3.69 | 3.64–3.74 | < .001 |

HR: Hazard ratio, CI: Confidence interval

All values were weighted and derived from the MHAS sampling weights.

*Male sex was used in the multinomial regression model.

model revealed a HR of 1.63 (95% CI: 1.63–1.64, p< 0.001) and the male sex yielded a HR of 3.69 (95% CI: 3.64–3.74, p<0.001).

## Discussion

Our study demonstrates that the proposed BHI is useful in determining the risk of developing PVD. It also showed that at a higher score the greater the risk of developing PVD and stroke in Mexican adults aged 50 years or older. It is important to underscore that the BHI model revealed a greater risk of PVD independently of age, sex, and educational level in the 3 diagnostic categories studied.

To the best of our knowledge, this is the first study that has measured the impact of CVRFS as an index, in Mexican older adults. Since low- and middle-income countries have a high cardiovascular disease burden and CVRFS associated with dementia and PVD are potentially modifiable, it is essential to investigate the predictive capabilities of a BHI adapted to the Mexican population [28, 29]. Given the existence of previous adaptations and to prove the performance of this risk factor set model, we modified the AHA Life's Simple 7 CVRFS according to the MHAS questionnaire and replaced "dietary habits" and "cholesterol levels" with a "history of myocardial infarction" and "self-reported depressive symptoms". Additionally, and considering that dementia has a multifactorial etiology, the sum of individual factors has been proven useful in the detection of those at risk [13]. Several modifiable risk factors such as midlife obesity, smoking, depression, and physical inactivity, among others, where found as pivotal in dementia prevention [30].

The AHA's Life's Simple 7 score has been previously implemented to identify cognitive and cerebrovascular outcomes in other populations. In a 10-year follow-up analysis of the Framingham Study, participants that successfully achieved the ideal goals of care of every CVRFS had a lower incidence of all-cause dementia, AD, stroke, and PVD [31]. It is particularly relevant that other studies have adapted the score according to their own population, their type of epidemiological questionnaires, and to gain a better understanding of their individual cognitive outcomes. An example of the latter is a Finnish study that used the CAIDE score information to modify the AHA Life's Simple 7 ideal goals, but they omitted the "diet information" and used the "history of treatment of diabetes during midlife" instead. The authors showed that maintaining these updated goals decreased the risk of developing dementia later in life [25].

Another study that used a modification of the original ideal goals included in the AHA Life's Simple 7 scale, is the Study on Global Ageing and Adult Health (SAGE). In its Chinese adaptation, instead of glucose and cholesterol levels, the authors included factors such as alcohol consumption and self-reported anxiety; they clearly reproduced previous findings in which maintaining these ideal goals was associated with better cognitive performance [32]. In a Hispanic sample, the AHA Life's Simple 7 was also used to establish its association with neurocognition, whereby a better score (improved health metrics) was associated with a higher neurocognitive function specially in the female sex. In this same study, a lower educational level was associated with a greater risk of dementia (worsened neurocognitive function) [33].

Previously, an analysis from the MHAS study showed that cognitive impairment was dependent on CVRFS such as diabetes mellitus, stroke, and depression, as well as other variables such as age and being female [34]. Another study showed that depression and diabetes mellitus were associated with mild cognitive impairment (MCI) but without memory compromise, while midlife hypertension and diabetes mellitus were associated with MCI in late life [35].

In our logistic regression model, in a similar way as in a previous study [5], advanced age was associated with the cognitively impaired and PVD groups. Also in our study, a higher educational attainment remained a protective factor in the dementia and stroke groups, as

reported by Mirza et al. [36]. Being male was also a strong risk factor in the vascular diagnostic categories, as shown in previous risk models [37, 38].

Our study shows that prior myocardial infarction (6.9%) and smoking (59.5%) were significantly associated with PVD. Heart disease is an independent risk factor for the development of VCI. In a study of post-menopausal women, those with heart disease (defined as self-reported vascular disease, atrial fibrillation, or heart failure) and after excluding subjects with a history of stroke, the risk of developing cognitive impairment had increased after an 8.4-year follow-up [39]. Smoking has been identified as a major risk factor for all-cause dementia and accounts for up to 5% of potentially preventable causes [10]. In a Japanese cohort of individuals aged ≥65 years, the authors showed a greater risk for all-cause dementia in active smokers in comparison with non-smokers [HR 2.28 (95% CI: 1.49–3.49, p<0.001)], AD [HR 1.98 (CI: 1.09–3.61, p <0.001)], and VaD [HR 2.88 (95% CI: 1.34–6.20, p <0.001)], with similar associations among midlife smokers [40].

Hypertension, a key modifiable risk factor, has already been proven to be a pivotal CVRFS [41]. In our study, hypertension was more frequent among participants with stroke, dementia, and PVD, which correlates with the physiopathology of PVD. As shown in rat models with dysfunction of the neurovascular unit, cerebral perfusion abnormalities could represent the earliest preclinical changes in dementia patients, even in neurodegenerative entities [42]. It has been demonstrated in longitudinal studies that high blood pressure in midlife and late life can increase the risk of cognitive decline and dementia, particularly vascular dementia [43].

Depression remained a statistically significant factor in the 3 diagnostic categories included in our study. It was, however, more prevalent among participants in the dementia group (55.8%). The presence of depression supports a bidirectional relationship with the pathophysiology of dementia, reflecting an inflammatory state with an increase in βA deposition and neurofibrillary tangle formation [44]. In PVD, magnetic resonance imaging in subjects with depression showed the contribution of white matter hyperintensities (WMH) and CVRFS to the development of cognitive impairment [45].

Surprisingly, diabetes mellitus was the least prevalent health variable in the PVD group (16.5%), while it was the most frequent among participants in the stroke group (38.8%). These results are interesting due to the existence of established and growing evidence linking diabetes and cognitive impairment, based on a recent metanalysis suggesting that even the presence of prediabetes and changes in glucose biomarkers predicted an increased risk of developing cognitive impairment [46]. Previously, in the MHAS study, a cross-sectional analysis of glycated hemoglobin (HbA1c) levels showed that poor diabetes control (>8%) was associated with a worse cognitive performance [47]. In VaD, the Vantaa study of Finnish elders aged ≥85 years and followed over 10 years, and whose brains were examined at autopsy, showed that individuals with diabetes were more likely to have a cerebral infarction [HR 1.88 (95% CI: 1.06–3.34, p <0.001)] [48].

Physical inactivity was most frequent in the PVD group (87.3%); this was expected given that patients with VaD have a higher disability burden [3]. Physical inactivity has been previously considered a main modifiable risk factor, with interventions leading to improved cognitive function among dementia patients. However, most interventional trials do not establish distinctions between VaD and neurodegenerative disease [49]. Nevertheless, there is proven evidence of its major contribution to the prevention of cognitive impairment in midlife since increasing physical activity decreases the incidence of VaD [HR 0.65, 95% CI: 0.49–0.87)] [50].

One of the strengths of our study was the analysis of the performance of the BHI over three years and to finding of a longitudinal association in the development of PVD. This becomes relevant when considering that in Latin Americans there is a greater burden of CVRFSs [28], while other well-established risk factor for the development of dementia, such as the presence

of apolipoprotein E-ε4, is less prevalent in Mexican subjects, suggesting that the focus should be directed to potentially modifiable risk factors [51]. The study also shows the existing contribution of multimorbidity detected with the BHI, as a proxy of both multimorbidity and brain health, since subjects that complied with the definition of dementia or stroke had a certain degree of disability, which translates into an increased need for specialized care [3]. The latter suggests that the BHI can be considered a potential prognostic, especially if we consider that in our sample, even the CU subjects had at least 2 CVRFS.

Finally, our study also has a few limitations. First, and considering the operational definitions of the diagnostic categories, data was collected with a questionnaire that does not permit the differentiation between IADLs that were lost because of a stroke, cognitive impairment, or other causes. Second, and what may be pivotal in this type of epidemiological studies, is the fact that surveys are not correlated with neuroimaging studies, thus decreasing the level of certainty of a VCI diagnosis; hence, according to the VASCOG [24] and VICCCS [7] criteria, only a "possible" diagnosis can be established. Furthermore, we cannot exclude the presence of cerebrovascular disease such as WMH and lacunar infarctions, which could lead to a misdiagnosis of non-vascular cognitive impairment [52]. Third, the cognitive assessment used in the MHAS does not allow us to identify a cognitive profile among the participants, such as a non-amnestic multiple domain profile, which could suggest a vascular etiology with more certainty [53]. Lastly, as a retrospective survey, loss of data must be accounted for (addressed by weighing of cases), especially considering that 2/3 of the PVD group were proxy respondents, which interferes with the ascertainment of certain covariables such as symptoms of depression.

## Conclusions

In this longitudinal study, with the use of a practical multifactorial BHI in the Mexican population, regardless of age and educational level, a higher score was associated with a 1.64-fold increase in the risk of developing PVD and a 1.75-fold increase in the risk of stroke; the index also allowed us to detect an increased risk in non-stroke dementia over the 3 years of follow-up.

## Supporting information

**S1 Table. Demographic characteristics of MHAS sample 2012 by diagnostic group.** (DOCX)

## Acknowledgments

All authors significantly contributed to the conception, design, analysis, and interpretation of the manuscript. Authors also contributed to the draft and revision of the content and approved the last version submitted for publication.

## Author Contributions

**Conceptualization:** Sara G. Aguilar-Navarro, Sara G. Yeverino-Castro, Silvia Mejía-Arango, Teresa Juárez-Cedillo, Alberto José Mimenza-Alvarado.

**Data curation:** Sara G. Yeverino-Castro, Rogelio Moctezuma, Teresa Juárez-Cedillo.

**Formal analysis:** Sara G. Aguilar-Navarro, Sara G. Yeverino-Castro, Silvia Mejía-Arango, Rogelio Moctezuma, Teresa Juárez-Cedillo, Alberto José Mimenza-Alvarado.

**Funding acquisition:** Sara G. Aguilar-Navarro, Sara G. Yeverino-Castro, Rogelio Moctezuma.

**Investigation:** Sara G. Aguilar-Navarro, Rogelio Moctezuma, Teresa Juárez-Cedillo.

**Methodology:** Sara G. Aguilar-Navarro, Sara G. Yeverino-Castro, Rogelio Moctezuma, Teresa Juárez-Cedillo, Alberto José Mimenza-Alvarado.

**Project administration:** Rogelio Moctezuma.

**Resources:** Rogelio Moctezuma.

**Software:** Sara G. Yeverino-Castro, Rogelio Moctezuma, Teresa Juárez-Cedillo.

**Supervision:** Silvia Mejía-Arango, Teresa Juárez-Cedillo.

**Validation:** Silvia Mejía-Arango, Teresa Juárez-Cedillo.

**Visualization:** Sara G. Yeverino-Castro, Silvia Mejía-Arango, Teresa Juárez-Cedillo.

**Writing – original draft:** Sara G. Aguilar-Navarro, Sara G. Yeverino-Castro, Silvia Mejía-Arango, Rogelio Moctezuma, Teresa Juárez-Cedillo, Alberto José Mimenza-Alvarado.

**Writing – review & editing:** Sara G. Aguilar-Navarro, Sara G. Yeverino-Castro, Silvia Mejía-Arango, Rogelio Moctezuma, Teresa Juárez-Cedillo, Alberto José Mimenza-Alvarado.

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
