## [Decision Letter · Decision Letter 0]

27 Dec 2023

PONE-D-23-19179Brain health index as a predictor of possible vascular dementia in the Mexican Health and Aging Study 2012-2015PLOS ONE

Dear Dr. Navarro

Thank you for submitting your manuscript to PLOS ONE. After careful consideration, we feel that it has merit but does not fully meet PLOS ONE’s publication criteria as it currently stands. Therefore, we invite you to submit a revised version of the manuscript that addresses the points raised during the review process.

We look forward to receiving your revised manuscript.

Kind regards,

Vikash Jaiswal, MD

Academic Editor

PLOS ONE

Journal Requirements:

Additional Editor Comments (if provided):

Dear Authors,

Kindly revise and add graphical abstract with aim, methods, results, conclusion.

Thanks

Reviewers' comments:

Reviewer's Responses to Questions

**Comments to the Author**

1. Is the manuscript technically sound, and do the data support the conclusions?

Reviewer #1: Yes

2. Has the statistical analysis been performed appropriately and rigorously? 

Reviewer #1: Yes

3. Have the authors made all data underlying the findings in their manuscript fully available?

Reviewer #1: Yes

4. Is the manuscript presented in an intelligible fashion and written in standard English?

Reviewer #1: Yes

5. Review Comments to the Author

Reviewer #1: The authors of this manuscript attempt to show the efficacy of a new brain health index proposal and to associate it with possible vascular dementia. The issue/topic is exciting, but some aspects need to be clarified and revised to make the article more robust and rigorous:

Overall comments:

1. The abstract objective and the main text must be consistent. Please clarify.

2. Specify the study design in the methods section

3. In the discussion section, it would be helpful to claim why the BHI was adapted for the Mexican population as the proposal in this paper.

Minor comments

1. Line 59, "a variety of" change for "various"

2. Line 60, "and accounts" for change for "accounting"

3. Line 64, "lead" change for "leads"

4. Line 119, "This study's protocol", please rephrase it. This is a manuscript.

5. Why is the INCMNSZ in Spanish and other institutions, as INEGI is in English?

5. To help the reader, it is better if the description of the table in lines 241 - 246 is in the same order as the table.

6. In Table 2, please use a * in the Sex word to help the reader see that the male sex was used for the multinominal regression model. The word multinominal is misspelled.

6. PLOS authors have the option to publish the peer review history of their article (what does this mean?). If published, this will include your full peer review and any attached files.

Reviewer #1: **Yes: **Samuel Gutierrez-Barreto

---

## [Author Response · Author response to Decision Letter 0]

14 Jan 2024

Response to reviewers

PLOS ONE Journal

Dear editor in chief, academic editor, and reviewer:

We are thankful for the thoughtful comments on our manuscript. Your valuable and insightful comments have led to possible improvements in the current version. The authors have carefully considered the comments and tried our best to address each one of them. Moreover, the authors welcome further constructive comments if any. 

In this letter, our responses are written in purple. We have included a marked-up copy of the manuscript where changes are highlighted in yellow.

In response to Journal’s additional requirements:

We have modified the manuscript’s, title (lines 2-3) and abstract section (lines 28-49) to meet PLOS One’s manuscript title, author, affiliations, and body formatting guidelines.

2. Funding Information and Financial Disclosure sections do not match.

We have corrected the mentioned sections and included a statement in the revised cover letter; “The authors declare that they have no conflicts of interest and that this research has not received any financial funding.”

3. Data availability statement.

We would like to change our availability statement as follows: “Data are available from The Mexican Health and Aging Study (MHAS) from www.mhasweb.org”. 

The manuscript’s methodology section (lines 118-119) includes an e-link to access the MHAS data and documentation, which are of public use and available at www.mhasweb.org. If specific data is required, please let us know.

4. Please review your reference list to ensure that it is complete and correct.

Thank you for your suggestion. We have carefully reviewed the reference list and corrected a mistake in line 295, where an author’s first name was misplaced. We have ensured this list is complete and correct. We did not include any retracted papers.

In response to Reviewer #1:

1. The abstract objective and the main text must be consistent. Please clarify. 

We are very thankful of your comment. We have homogenized the concepts included in the abstract (lines 33-35) for it to be consistent with the objective described in the main text (lines 101-102). 

2. Specify the study design in the methods section. 

We are grateful you have identified this. We have addressed this issue in line 105, 109-112.

3. In the discussion section, it would be helpful to claim why the BHI was adapted for the Mexican population as the proposal in this paper.

We are very thankful of your suggestion. We have added a sentence as to why it is important to adapt the BHI to the Mexican population in lines 254-257.

4. Suggestions in lines 59, 60, 64, and 119.

Thank you for your suggestions. In this revised version of the manuscript, we have addressed these issues in lines 52, 54, 57, and 113, respectively.

5. Why is the INCMNSZ in Spanish and other institutions, as INEGI is in English.

Thank you for identifying this. We have included this Institution’s name (INCMNSZ) in English (line 114).

6. To help the reader, it is better if the description of the table in lines 241 - 246 is in the same order as the table.

We are very thankful of your comment. We have updated Table’s 2 description in lines 234-238.

7. In Table 2, please use a * in the Sex word to help the reader see that the male sex was used for the multinominal regression model. The word multinominal is misspelled.

We appreciate your comment and have corrected this issue in Table 2 (lines 241-245). 

Sincerely, 

Sara Gloria Aguilar Navarro, PhD 

sara.aguilarn@incmnsz.mx

Professor, Deparment of Geriatric Medicine,

Instituto Nacional de Ciencias Médicas y Nutrición Salvador Zubirán, Tlalpan, Mexico City, Mexico

---

## [Decision Letter · Decision Letter 1]

9 May 2024

Brain health index as a predictor of possible vascular dementia in the Mexican Health and Aging Study 2012-2015

PONE-D-23-19179R1

Dear Dr. Aguilar-Navarro,

We’re pleased to inform you that your manuscript has been judged scientifically suitable for publication and will be formally accepted for publication once it meets all outstanding technical requirements.

Kind regards,

Victor Manuel Mendoza-Nuñez, PhD

Academic Editor

PLOS ONE

---

## [Editor Report · Acceptance letter]

13 May 2024

PONE-D-23-19179R1 

PLOS ONE

Dear Dr. Aguilar-Navarro, 

I'm pleased to inform you that your manuscript has been deemed suitable for publication in PLOS ONE. Congratulations! Your manuscript is now being handed over to our production team.

Kind regards, 

on behalf of

Dr. Victor Manuel Mendoza-Nuñez 

Academic Editor

PLOS ONE